# Effects of Physical and Chemical Factors on the Structure of Gluten, Gliadins and Glutenins as Studied with Spectroscopic Methods

**DOI:** 10.3390/molecules26020508

**Published:** 2021-01-19

**Authors:** Konrad Kłosok, Renata Welc, Emilia Fornal, Agnieszka Nawrocka

**Affiliations:** 1Institute of Agrophysics, Polish Academy of Sciences, Doświadczalna 4, 20-290 Lublin, Poland; k.klosok@ipan.lublin.pl (K.K.); r.welc@ipan.lublin.pl (R.W.); 2Department of Pathophysiology, Medical University of Lublin, Jaczewskiego 8b, 20-090 Lublin, Poland; emilia.fornal@umlub.pl

**Keywords:** gluten, gliadins, glutenins, infrared spectroscopy, Raman spectroscopy, proteins structure, dietary fibre preparations, hydrocolloids

## Abstract

This review presents applications of spectroscopic methods, infrared and Raman spectroscopies in the studies of the structure of gluten network and gluten proteins (gliadins and glutenins). Both methods provide complimentary information on the secondary and tertiary structure of the proteins including analysis of amide I and III bands, conformation of disulphide bridges, behaviour of tyrosine and tryptophan residues, and water populations. Changes in the gluten structure can be studied as an effect of dough mixing in different conditions (e.g., hydration level, temperature), dough freezing and frozen storage as well as addition of different compounds to the dough (e.g., dough improvers, dietary fibre preparations, polysaccharides and polyphenols). Additionally, effect of above mentioned factors can be determined in a common wheat dough, model dough (prepared from reconstituted flour containing only wheat starch and wheat gluten), gluten dough (lack of starch), and in gliadins and glutenins. The samples were studied in the hydrated state, in the form of powder, film or in solution. Analysis of the studies presented in this review indicates that an adequate amount of water is a critical factor affecting gluten structure.

## 1. Introduction

Wheat bread is a staple food in the Western diet. From the technological point of view, its quality depends on flour quality and bread-making procedure [1]. Biochemically, the bread quality is strictly related to the structure of gluten and gluten proteins (gliadins and glutenins), which may depend on different factors such as the amount of water added to dough, temperature and time of dough mixing, and type of compounds added to dough. The structure and structural changes of the proteins can be studied directly or indirectly using different methods. The direct methods include X-ray diffraction, infrared and Raman spectroscopy, and circular dichroism. Whereas indirect techniques provide information among others by determination of aggregates formation (electrophoresis, size exclusion HPLC, RP-HPLC, LC-MS), changes in thermal properties (thermogravimetry, differential scanning calorimetry), and changes in water populations and pattern of hydrogen bonding (NMR) [2]. All the mentioned techniques are used to determine gluten structure. However, most of them require time consuming and expensive preparation of the samples, expensive and sophisticated equipment and software to analyse data or the samples are destroyed during the tests. Vibrational techniques like infrared spectroscopy and Raman spectroscopy seem to be an adequate analytical approach to study structure of gluten proteins since they do not require any expensive reagents or equipment.

## 2. Biochemistry and Structure of Gluten Network and Gluten Proteins

Gluten is a continuous and viscoelastic network that is formed during mixing process and is responsible for dough as well as bread quality. Gluten contains two types of proteins—alcohol-soluble gliadins and alcohol-insoluble glutenins—which are characterized by high content of glutamine (35%), glycine (20%) and proline (10%) and low contents of amino acids with charged groups [3]. Generally, gluten can be regarded as a ‘two-component glue’, in which gliadins act as a plasticizer for glutenins. From the technological point of view, gliadins contribute to the viscosity and extensibility of the dough, whereas glutenins are responsible for dough strength and elasticity [4].

Gliadins are monomeric proteins that can be classified into three groups (α-/β-, γ- and ω-gliadins) depending on the protein mobility at low pH in gel electrophoresis. The α-/β- and γ-gliadins are similar in terms of molecular weight (30–35 kDa) and amount of disulphide bonds (three and four intrachain S=S bonds, respectively). Some of the gliadins have an odd number of cysteines that is proposed to act as terminator of glutenin polymerization [4]. Molecular weight of ω-gliadins is 44–88 kDa and they do not form disulphide bridges because they lack cysteine in their amino acid sequence [5]. They contain repetitive sequences rich in glutamine and proline e.g., PQQPFPQQ. As for the secondary structure, α-/β- and γ-gliadins contain mainly α-helices and β-sheets, whereas β-turns dominate in ω-gliadins [6].

Glutenins are polymeric proteins insoluble in alcohol solutions whose molecular weight ranges from 500 kDa to 10 MDa. Glutenins consist of two subunits differing in molecular weight—low (LMW) and high molecular weight (HMW). The glutenin subunits (GS) are characterized by alcohol solubility similar to gliadins after treating them with reducing agent caused cleavage of disulphide bridges. The LMW-GS are similar to α-/β- and γ-gliadins in molecular weight (30–45 kDa) as well as primary and secondary structures. In contrast to the gliadins, the LMW-GS also participate in formation of interchain S=S bonds that allow formation of branched glutenin network. The HMW-GS are grouped into two types, x- and y-type, differing in molecular weight (83–88 kDa and 67–74 kDa, respectively). Both types of HMW-GS have a typical three-domain structure containing N- and C- termini and central domain. The domains differ in the secondary structure. The N- and C- termini are rich in α-helices, whereas the central domain contains overlapping reverse turns that form a β-spiral structure [7]. Therefore, the HMW-GS can form a rigid rod-like conformation [8]. Additionally, some portion of glutenins can form a SDS-insoluble gel containing large glutenin aggregates that is known as glutenin macropolymer (GMP). The GMP quantity affects strongly the elastic properties of dough and bread loaf volume [9,10].

Gliadins and glutenins interact with each other by formation of covalent and non-covalent bonds to obtain gluten network during the dough mixing process. One of the crucial covalent bonds present in the gluten network are disulphide bridges (S=S bonds), which are formed by cysteine residues from the same protein complex (intrachain S=S bonds) or different protein complexes (interchain S=S bonds). As it was mentioned above, gliadins form only intrachain disulphide bridges, whereas glutenins participate in intra- and interchain S=S bonds. Therefore, gliadins with intramolecular S=S bonds are not involved in the SH-SS interchange, which takes place during dough mixing process. While SH-SS interchange concerns intermolecular disulphide bridges in the glutenins and thus both glutenin subunits can act as a chain extender [11]. Another covalent bonds formed during dough mixing are tyrosine—tyrosine crosslinks between gluten polypeptide chains [12,13], and tyrosine—dehydroferulic acid bonds between gluten proteins and arabinoxylans [14]. However, these two crosslinks are rarely formed [15]. The gluten network is also formed due to the presence of non-covalent interactions such as hydrogen bonds, ionic bonds, hydrophobic bonds and Van der Waals forces [4].

**Table 1 molecules-26-00508-t001:** Contents of particular secondary structures in different types of wheat gluten determined with application of infrared spectroscopy.

Type of Gluten	Content of Protein Secondary Structures		
	α-Helix	β-Sheets	β-Turns
Native [16]	51%	36%	13%
Commercially available [17,18]	34% 45%	49% 47%	17% 8%
Hydrated solid state [19]	31%	28%	27%

Spectroscopic methods can be used to study secondary structure of native and commercially available gluten. Additionally, the gluten samples can be studied in different states—powder and hydrated gluten dough. Also different spectroscopic methods can be applied depending on what kind of structural information a scientist wants to obtain. Contents of particular secondary structures in different types of wheat gluten determined with application of infrared spectroscopy are presented in Table 1. Analysis of the structure of the particular gluten proteins were also found in the literature. Structure of two gliadin fractions in solution (ω- and γ_46_-gliadins) were determined [19]. γ_46_-gliadins contained more α-helices (36%), whereas β-sheets (31%) and β-turns (31%) dominated in ω-gliadins. The results indicated that extended β-sheets are located in the repetitive domains of gluten complex and are responsible for viscoelastic properties of gluten. Structure of the dried wheat gluten, gliadins and glutenins, which were extracted from the dried wheat gluten, were studied by Rasheed et al. [20]. FT-IR spectra of the studied samples showed dominance of α-helical structures. These studies included also prediction of the structural model of gliadins and glutenins. Structure of the HMW glutenin subunit as a whole and only its central domain without N- and C- termini was studied by Gilbert et al. [21]. Analysis of the FT-IR spectra showed that central repetitive domain formed β-turns and intermolecular β-sheets in the dry and hydrated solid state. In contrast, analysis of the whole subunits showed presence of the α-helix in the C- and N- termini. Additionally, the spectroscopic results indicated that N- and C- termini of the HMW subunits promote formation of the β-sheets between repetitive domains. In order to determine exact structure of the gluten proteins, the model peptides of specified amino acid sequence and length, which mimic the repetitive domain of HMW glutenin subunits, were studied. Cyclic and linear peptides differing in the amount and sequence of amino acids were synthesised [22] and their structure in the solution and dry state were studied with application of FT-IR spectroscopy [23]. The results showed that both linear and cyclic peptides adopt β-turn structure. Depending on the amino acid sequence they may form two types of β-turns (type I and type II). The β-turn types differ in the amino acids sequence preferences [24]. Type I β-turns were formed in peptides containing glutamine, proline and glycine, whereas presence of tyrosine, threonine, serine, glutamine and proline induced formation of type II β-turns [22]. FT-IR spectra of the peptides contain two bands at 1638 and 1642 cm^−1^ that can be assigned to type II and type I β-turns, respectively. The results showed that the peptides in solution may adopt both types of β-turns, however, only β-turns of type II were observed in the dry state. The Authors claimed that the repetitive domain of the dry HMW GS may contain a continuous series of β-turns, which organize into β-sheet structures [23]. Similar behaviour was observed in the peptides mimicking the HMW subunits by Feeney et al. [25]. The studied peptides contained two repetitive motifs PGQGQQ + GYYPTSLQQ, which formed peptide chains consisted of 21, 45, 110 and 203 residues. At low water content β-sheets dominate in the peptides structure. An increase in the hydration level led to an increase in the β-turn content at the expense of β-sheets for the short peptides, whereas formation of intermolecular β-sheets was observed for the long peptides. The FT-IR spectra of the long peptides showed two bands at 1616 and 1685 cm^−1^, that are characteristic for the intermolecular β-sheets in aggregated proteins. These results may confirm the ‘loop and train’ model of gluten proposed by Belton [26]. Long peptides characterized by perfect repetition in amino acid sequence would participate in formation of train regions observed as intermolecular β-sheets. While peptides with differences in the amino acid sequence would form loops observed as β-turns and hydrated extended structures.

FT-IR spectroscopy has been used the most often to determine secondary structure of the gluten proteins. However, near-infrared (NIR) spectroscopy also was used to do it. Structure of the gluten in the powder and hydrated form were studied by Bruun et al. [27,28], respectively. Spectral data combined with chemometrics allowed to assign NIR bands to secondary structures. Bands at 2056, 2172, 2239, 2289 and 2343 nm were assigned to α-helix, 2205, 2264, and 2313 nm to β-sheets, and 2265 nm to random coils.

Analysis of the Raman spectra provides more detailed information about secondary structure of gluten proteins comparing to infrared spectra. Studies of Nawrocka et al. [29] showed that gluten network contained more than one type of β-sheets (parallel β-sheets, 5%) such as pseudo-β-sheets (7%) and antiparallel-β-sheets (16%). Additionally, it was determined α-helix as a dominant structure (54%), β-turns (5%), and some amount of aggregates (ca. 10%).

## 3. Spectroscopic Methods Used in the Study of Gluten Structure

Two spectroscopic methods, infrared spectroscopy and Raman spectroscopy, are used the most often to determine structure of gluten network as well as gliadins and glutenins. In the case of infrared spectroscopy, Fourier transform infrared spectroscopy (FT-IR) with attenuated total reflection (ATR) is used the most often. To study structure of the gluten proteins also Fourier transform Raman spectroscopy (FT-Raman) is used. Raman spectrophotometer is equipped with infrared Nd:YAG laser with λ = 1064 nm.

The infrared and Raman spectroscopies provide different but complimentary information about the proteins structure. Secondary structure can be determined with application of both method by analysis of amide I band (1570–1720 cm^−1^). Assignments of particular secondary structures in the amide I band are presented in Table 2. Amide III band in the FT-IR spectrum is also used to determine secondary structure of gluten proteins due to lack of water oscillations in this spectral region [30]. Amide III band can be divided into four spectral regions that can be assigned to the following secondary structures: β-sheets (1200–1250 cm^−1^), random coils (1250–1270 cm^−1^), β-turns (1270–1295 cm^−1^), and α-helices (1295–1330 cm^−1^) [31]. Additionally, analysis of the β-sheets region in the amide III band can provide information about two types of hydrogen bonds (type I and type II) [32] formed during dough mixing. According to Nawrocka et al. [33], the type I hydrogen bonds (–HN···O=C–) with a band at ca. 1230 cm^−1^ can be formed between polypeptide chains in the gluten network leading to its aggregation, whereas H bonds of type II (–HN···O, ether bond) with a band at ca. 1220 cm^−1^ can arise as a result of interactions between gluten proteins and different compounds added to the dough e.g., polysaccharides, polyphenols. Amide II band can be also used to determine secondary structure of the proteins. However, it is less reliable for this purpose comparing to amide I and amide III bands. It can be applied to determine changes in the proteins hydration because increase in the dough hydration level causes shift of the amide II band toward higher wavenumbers [34,35]. It is a well-known fact that water is necessary to obtain wheat dough characterized by appropriate mechanical properties. Water populations present in the gluten network can be determined by the analysis of the spectral region assigned to OH stretching (2800–4000 cm^−1^) in the FT-IR spectra. Water molecules interact with gluten polypeptide chains by strong and weak hydrogen bonds with characteristic bands at ca. 3055 and 3190 cm^−1^, respectively [36]. Additionally, other bands in the OH stretching region were assigned to water molecules participating in two hydrogen bonds (ca. 3280 cm^−1^) [37], small hydrogen-bonded water clusters (ca. 3370) [36] and free water (ca. 3650 cm^−1^) [37].

The analysis of the Raman spectra provides information about conformation of disulphide bridges, and behaviour of two aromatic amino acids—tyrosine and tryptophan [38]. The spectral region 490–550 cm^−1^ is connected with three conformations of disulphide bridges: *gauche-gauche-gauche* (g-g-g), *trans-gauche-gauche* (t-g-g), and *trans-gauche-trans* (t-g-t). According to Sugeta [39], each conformation (g-g-g, t-g-g, and t-g-t) can be assigned to the following maxima at ca. 505, 520, and 530 cm^−1^, respectively. Additionally, the Raman spectra can show two bands at ca. 515 and 540 cm^−1^ that are related to intrachain disulphide bonds in the t-g-g and t-g-t conformations, respectively [40,41]. Gluten contains three aromatic amino acids—phenylalanine (PHE), tyrosine (TYR) and tryptophan (TRP). The PHE band at ca. 1003 cm^−1^ is commonly used to normalize Raman spectra of gluten network because the intensity and location of the PHE band are not sensitive to protein conformation or to the microenvironment [42]. Tyrosine residues, which occur periodically throughout of the length of gluten proteins, show two band at 830 and 850 cm^−1^ in the Raman spectrum. Ratio of these bands intensities I(850)/I(830) (tyrosine doublet) is known as a good indicator of hydrogen bonding of the phenolic hydroxyl group. A decrease in tyrosine doublet suggests buriedness of TYR residues inside protein complex and formation of intramolecular hydrogen bonds. In contrary, an increase in the ratio value indicates exposition of the TYR residues on the surface of protein complex and interactions with different additives [43]. Tyrosine doublet takes specific values that indicate behaviour of TYR residues. According to Siamwiza et al. [44], the ratio value varied from 0.90 to 1.43 for ‘normal’ tyrosine, in which OH groups serves as both a donor and an acceptor of proton in H bond. The values from the range 0.30–0.90 indicate formation of strong hydrogen bonds in which TYR residues act as proton donor. If the ratio is higher than 1.50, the TYR residues serve as a proton acceptor in the hydrogen bond. A Raman spectrum of proteins show a few bands connected with oscillations of the TRP residues, e.g., tryptophan doublet I(1360)/I(1340) determined for proteins in solution [45], bands at 760 and 1554 cm^−1^ arising from indole ring vibrations [46]. In the case of gluten proteins, the band at 760 cm^−1^ is the most appropriate, and change in its intensity provides information about hydrophobicity of the indole ring [47] A decrease in the band intensity indicates that TRP residues are exposed on the surface of protein complex, whereas burriedness of the TRP inside hydrophobic environment of the protein complex is shown by increase of the band intensity. Behaviour of the TRP residues also can be studied with application of fluorescence spectroscopy since TRP exhibits a broad fluorescence band at ca. 350 nm [48,49].

## 4. Factors Affecting Structure of Gluten Network, Gliadins and Glutenins

Structure of the gluten network and particular gluten proteins changes during all the processes that a wheat flour and wheat dough undergoes such as dough mixing, mechanical deformation of the dough, dough freezing and frozen storage, addition of flour improvers (e.g., emulsifiers), supplementation of the dough with different chemical compounds (e.g., dietary fibre preparations, polysaccharides including hydrocolloids, polyphenols).

### 4.1. Dough Mixing

Viscoelastic gluten network is formed during dough mixing as a result of hydration, depolymerisation and re-polymerisation processes [67]. Chemical changes in the gluten network during dough mixing have been studied with application of Fourier transform infrared spectroscopy (FT-IR) with analysis of amide III band. This band was chosen to study secondary structure of hydrated dough due to lack of water oscillations. Ait Kaddour et al. [68] observed changes in the secondary structure of gluten during 20-min mixing as increase or decrease in the band intensities corresponding to the secondary structures in the amide III band. Increasing mixing time led to increase in the amount of α-helix, β-sheets and β-turns. The results indicated formation of more ordered gluten structure during mixing. Similar results obtained Seabourn et al. [69] who analysed second derivative band area instead of band intensity. Structural modifications in the gluten proteins were studied during mixing of strong and weak wheat doughs [70]. Weak and strong wheat doughs were prepared from hard and soft wheat varieties, respectively. FT-IR spectra showed formation of β-sheets during dough mixing regardless of the dough type. However, the mechanism of β-sheets formation differs for both dough types. The results indicated that disulphide bridges can induce formation of β-sheets in the strong doughs, whereas hydrophobic interactions play an important role in the weak doughs. Attempts were also made to assign the amide bands from MIR spectra to bands from NIR spectra. Both kinds of spectra were registered during dough mixing [71]. Analysis of the raw NIR and MIR spectra demonstrated that the following NIR bands (1100–1167), (1235–1268), (1508–1834), (1928–1952), and (2200–2231) nm are highly correlated to amide bands in the MIR spectra. However, analysis of the second derivatives, which is regarded as more specific by the Authors, showed correlation with only three NIR bands (1189–1216), (1351–1474), and (2280–2325) nm.

During dough mixing water act as a plasticizer and strongly affect gluten structure [72]. In other words, the amount of water added to the dough (dough hydration level, water absorption) determined formation of particular secondary structures. Spectroscopic studies of Almutawah et al. [73] showed changes in the gluten structure as an effect of increase in the hydration level from 0% to 63%. Dry gluten contains a large amount of unordered structures with a band at 1645 cm^−1^ in the FT-IR spectrum. This band split into two bands at 1630 and 1650 cm^−1^ assigned to β-sheets and α-helices, respectively, as a result of hydration. Further increase in the hydration level leads to formation of extended hydrated chains and β-turns. Similar results concerning structural changes were observed for dough at 90% water absorption [74], HMW glutenin subunits [34], and ω-gliadins [35]. The hydration of the gluten network during the dough mixing also can be studied by using NIR spectroscopy. Studies of Wesley et al. [75] and Alava et al. [76] showed that bands at 1160 and 1200 nm were attributed to changes in water and proteins, respectively.

Another parameter crucial for the adequate course of the dough mixing process is temperature. Effect of the temperature (4, 15 and 30 °C) on the gluten structure in strong and weak wheat doughs during mixing was studied by Quayson et al. [77]. Strong and weak wheat doughs were obtained from hard and soft wheat flours, respectively. Analysis of the FT-IR spectra of both doughs mixing at 30 °C showed the dominance of β-sheets and β-turns. However, decrease in the temperature induced formation of different structural changes in the studied doughs. In the strong dough, a slight increase in the β-sheet content were observed, whereas amount of β-turns, α-helices and random coils remained constant. In the weak doughs, amount of β-turns significantly increased at the expense of β-sheets. Contents of α-helices and random coils remained constant. These changes can arise from the kind of bonds which participate in formation of gluten network during dough mixing. The interactions driven the formation of gluten network at 30 °C are mainly covalent, while in the case of lower temperatures (4 and 15 °C) hydrophobic interactions play a major role. The mixing temperature alone has a little effect on gluten structure but it affected strongly the way gluten network interact with water. Georget and Belton [50] studied effect of temperature range 25–85 °C on the gluten conditioned at three water contents 0%, 13% and 47%. In the case of 0% hydrated gluten, heating did not induce any structural changes. Analysis of the FT-IR spectra showed significant changes caused by a temperature increase in the spectral region 1610–1630 cm^−1^ at 13% and 47% water content. Within this spectral region two bands were observed at 1613 and 1629 cm^−1^ that can be assigned to β-sheets with strong and weak hydrogen bonds, respectively. Additionally, heating induced formation of weakly hydrogen bonded β-sheets at the 13% water content, whereas amount of β-sheets with strong hydrogen bonds remained constant. Increase in both the temperature and water content to 47% led to formation of weakly H bonded β-sheets and extended hydrated chains. Simultaneously, heating caused appearance of β-turns and loss in α-helices. The cooling process showed that the structural changes are totally or partially reversible at 13% and 47% water content, respectively.

### 4.2. Mechanical Deformation of the Dough

One of the parameters describing dough quality is dough extensibility (resistance to extension) that is determined by stretching dough. Combination of rheological and spectroscopic studies showed that dough extensibility can be connected with structure of gluten network [59]. ATR-FT-IR studies of van Velzen et al. [78] showed that dough stretching resulted in increase in the content of extended β-sheets with simultaneous considerable decrease in the α-helix content. Slightly different results obtained Wellner et al. [79], who studied changes in gluten structure during a few cycles of extension and relaxation of gluten isolated from developing grain with application of dynamic FT-IR. The extensions caused increase in β-sheets at the expense of β-turns and α-helices. Part of the β-sheets reverted to random coils during relaxation. The authors claimed that the observed structural changes during extension concerns mainly HMW GS and are consistent with ‘loop and train’ model proposed by Belton [26].

### 4.3. Dough Freezing and Frozen Storage

Freezing process and frozen storage are used in the bakery to extend shelf life and preserve freshness of the product. Structural changes induced in the gluten network during process of dough freezing were determined by Meziani et al. [80]. Analysis of the amide III band in the FT-IR spectra indicated that freezing induced the protein aggregation which was observed as an increase in the amount of extended β-sheets with simultaneous decrease in α-helices content. The β-turns and random coils were not affected by the freezing. It was assumed that freezing caused partial unfolding of the protein complex with simultaneous formation of weak hydrogen bonds. Similar structural changes in the gluten structure were observed during the first four weeks of dough frozen storage [81]. After four weeks, a slight decrease in the content of all secondary structures were determined. This phenomenon can be connected with the water distribution in the dough during ice recrystallization. Influence of 60-day frozen storage on the structure of gluten-, gliadin-, and glutenin-rich fractions were studied by Wang et al. [62,82]. Analysis of the FT-IR spectra in the amide I band showed that β-sheet content did not depend on the duration of the frozen storage, whereas α-helices, antiparallel-β-sheets and β-turns were sensitive to this parameter. In the gluten- and gliadin-rich fractions, formation of the antiparallel-β-sheets and β-turns at the expense of α-helices were determined. However, no structural changes were observed in the glutenin-rich fraction during the storage. Similar to Meziani et al. [81], the observed changes in the protein structure were assigned to protein aggregation and thus to gluten deterioration. Protein aggregation was also indicated as a result of frozen storage of dehydrated gluten by Zhao et al. [60]. Similar to the studies described above, it was observed increase in the hydrogen bonded β-sheets with simultaneous decrease in the α-helices content during the 120-day storage. However, the frozen storage did not affect secondary structure of the 60% hydrated gluten.

### 4.4. Dough Improvement

To improve dough quality different compounds like sodium and potassium salts or emulsifiers can be added. Salts belonging to Hofmeister series like NaCl, NaBr and NaI are known to worsen rheological properties of wheat dough, and thus affect the gluten structure in the negative way [83]. Wellner et al. [84] studied effect of these salts on secondary structure of gluten network. Incubation of gluten samples with low levels of NaCl and NaBr led to a small increase in content of intermolecular β-sheets with simultaneous decrease in β-turns that indicated proteins aggregation. Further increase in NaCl content did not affect β-sheets, whereas NaBr decreased β-sheets content. Addition of NaI, regardless of its concentration, caused increase in the β-turns at the expense of intra- and intermolecular β-sheets. Contents of α-helix and random coils were affected slightly by the studied salts. Slightly different changes in the gluten structure was observed after addition of NaCl, KCl and cysteine to gluten dough by Mejri et al. [85] Analysis of FT-IR spectra showed increase in β-turns and extended structures at the expense of α-helix. These results indicated enhancement of water-protein interactions by reduction of protein-protein interactions which were observed as an increase in gluten solubility.

Influence of emulsifiers on the secondary and tertiary structure of the gluten proteins in wheat dough was studied by Ferrer et al. [40] and Gomez et al. [86]. Ferrer et al. [40] modified gluten proteins by different concentrations of sodium stearoyllactylate (SSL) (0.25, 0.5 and 1% *w*/*w*) and analysed its structure using FT-Raman technique. An increase in the intensity of amide I band was observed after addition of the emulsifier and the highest intensity was detected for 0.25% concentration. Detailed analysis of the amide I band showed increase in the α-helix content that were accompanied by a decrease in the amount of β-sheets. FT-Raman spectra provide also information about disulphide bridges. Analysis of the spectral region connected with S=S bonds show that only 0.25% addition of SSL resulted in the appearance of S=S bonds in the t-g-t conformation. Additionally, changes in the microenvironment of TYR and TRP were determined. Tyrosine doublet ratio increased in the sample containing 1% of SSL, whereas decreased for samples with 0.5% and 0.25% SSL content. It indicated exposition at the protein complex surface and burriedness in the protein complex of the TYR residues, respectively. TRP residues buried inside the protein complex after addition of 0.25% of SSL. Similar changes in the gluten structure after SSL addition to the dough were observed by Gomez et al. [86]. Additionally, Gomez et al. [86] studied also another emulsifier—diacetyl tartaric acid esters of monoglycerides (DATEM). In accordance with studies of Ferrer et al. [40], addition of the DATEM caused increase in the α-helix content with simultaneous reduction in the number of β-turns and β-sheets, and an increase in antiparallel-β-sheets content was observed. In the case of conformation of S=S bonds and TRP behaviour, DATEM caused similar changes to SSL. However, differences were observed in behaviour of TYR residues. All gluten-DATEM samples were characterized by lower I(850)/I(830) values compared to the control sample, regardless of the emulsifier concentration. In contrast to SSL, DATEM caused a protein folding without exposition of the tyrosine groups. Gluten modified with the mixture of SSL+DATEM showed a trend similar to that of gluten-SSL samples. To conclude, high concentrations of SSL (0.5% and 1%) and DATEM (1%) resulted in formation of more disordered protein structures.

### 4.5. Dough Supplementation

#### 4.5.1. Dietary Fibre Preparations (DFP)

Dietary fibre preparations (DFP) are added to wheat bread to increase the dietary fibre intake, since bread is a principal component of western diets and thus it seems to be a convenient way to deliver nutritional compounds (e.g., polysaccharides, polyphenols) to human organism. However, supplementation of a wheat bread with DFP leads to reduction of the bread quality that is strictly connected with the gluten structure [3].

Conformational changes in the gluten proteins in the dough prepared from a common wheat flour after 6% addition of apple-cranberry, cacao, carob and oat DFP were analysed by Nawrocka et al. [59] with application of FT-Raman spectroscopy. DFP caused an increase in the number of α-helix and aggregates with simultaneous decrease in the amount of β-structures. Additionally, new bands at 1622 and 1616 cm^−1^ were detected. The appearance of these bands indicated presence of β-sheets with strong, intermolecular hydrogen bonds, characteristic for aggregated proteins complexes in which antiparallel-β-sheets (pseudo-β-sheets) are created. Analysis of disulphide bridges region showed that addition of the DFP resulted in significant decrease in the number of S=S bonds in g-g-g conformation and an increase in the number of the bonds in t-g-g and t-g-t conformations. The tyrosine doublet decreased after apple-cranberry, carob and cacao fibres addition. The opposite effect, an increase in the ratio was observed for oat fibre. The results indicated greater burial of tyrosine residues and folding of gluten proteins as a result of H-O···H bonds formation in the case of all dietary fibres except for oat. The exposition of the tyrosine groups, observed after oat addition can be associated with its chemical structure, especially with the availability of the hydroxyl groups to interact with tyrosine residues in the gluten network. The results indicated aggregation of the gluten proteins as a result of bread dough supplementation with the studied DFP.

The physical mechanisms by which wheat bran affect the properties of gluten structure in the whole grain dough was studied by Bock et al. [87]. They used ATR-FT-IR technique to monitor the state of water and secondary structure of gluten proteins in wheat bran-supplemented doughs containing 35–50% moisture and 0–10% bran. Analysis of the OH stretching region of the control dough revealed the appearance of two peaks associated with two water populations: monomeric, non-hydrogen bonded water and water strongly hydrogen bonded to the gluten network. The population of monomeric water increased with increasing dough moisture. After addition of wheat bran, redistribution of water in the flour and bran dough system as well as reduction in monomeric water was observed. Amide I band studies showed that secondary structure of gluten in the control dough contained mainly β-sheets but the amount of β-sheets and unordered structures was greater than β-turns and α-helices. An increase in moisture of the control dough resulted in decrease in the number of unordered and α-helical structures with simultaneous increase in the amount of β-turns. Supplementation of the dough with bran induced an increase in β-sheet content with a simultaneous reduction in β-turn content. In general, supplementation of the wheat dough with wheat bran caused water redistribution among dough components and therefore promotes partial dehydration of gluten proteins and transformation of β-turns into β-sheets.

A common wheat flour contains native fibre substances that may interfere interactions between gluten network and DFP. Therefore, a model flour, reconstituted from wheat starch and wheat gluten in a constant weight ratio (80:15 *w*/*w*), is often used in this kind of studies, nowadays. Structural changes in the gluten network obtained from model dough can be studied with application of FT-Raman and FT-IR spectroscopies. Gluten samples from model dough supplemented with seven DFP in the concentration of 3–18% were investigated by Nawrocka et al. [88] with application of FT-Raman spectroscopy. Analysis of the difference spectra in the amide I band showed that all dietary fibres except oat induced similar changes in the secondary structure of gluten proteins. The most noticeable differences were detected in the regions associated with hydrogen bonded β-sheets and β-turns. The DFP also affected the conformations of disulphide bridges. The rise in the fibre concentration resulted in an increase in the number of S=S bonds in the stable g-g-g conformation in the case of cacao and decrease for the rest of fibres except for oat. The results indicated that increasing content of DFP in the case of carrot and carob induced the cleavage of the intrachain disulphide bonds. The opposite effect, the appearance of S=S intrachain bonds, was observed after addition of the higher concentration of cranberry and flax preparations. Conversion of the disulphide bridges from g-g-g conformation into less stable t-g-g and t-g-t conformations could result in the gluten proteins aggregation. The analysis of TYR and TRP behaviour also indicated gluten proteins aggregation. Additionally, Nawrocka et al. [29] obtained similar results in the research concerning interaction between DFP in 6% concentration and gluten proteins in the model flour. These studies indicated that the aggregation of gluten proteins can be the result of competition for water between gluten and DFP during dough mixing [89,90].

Mechanism of interactions between gluten network and the DFP were also studied with application of FT-IR spectroscopy [18]. Analysis of amide I and amide III bands confirmed that the addition of DFP resulted in aggregation of gluten proteins. Aggregated structures contained mainly hydrogen bonded β-sheets and antiparallel-β-sheets formed as a result of interactions between chains of gluten proteins or polysaccharides chains and gluten proteins. The biggest increment of aggregated β-sheets was induced by an increase in the fibre content from 3% to 6%. Moreover, the studies showed that DFP addition induced changes in water populations in the model dough. Changes observed in the OH stretching region as a result of DFP addition suggested that part of water molecules, which are involved in formation of weak hydrogen bonds with proteins in the control sample, can also participate in formation of weak hydrogen bonds with polysaccharide chains. The rest of water molecules induce formation of strong H-bonds with the gluten proteins which can lead to stiffening of the gluten network. However, the results suggested that strong hydrogen bonds are necessary to form gluten network of adequate mechanical properties. The FT-IR spectroscopy was also used to investigate effect of wheat dietary fibre and ferulic acid on the gluten proteins aggregation [91]. The spectra in the amide I band showed that the gluten secondary structure in all systems was dominated by β-sheets, which are considered as the most stable protein conformation. In the case of samples with ferulic acid combined with dietary fibre, a decrease in the number of β-sheets was observed as the amount of acid and fibre increased.

Pomaces obtained after cold pressing oil production can be also considered as DFP, since they are rich in dietary fibre, polyphenols and fatty acids. Effect of the supplementation of the model dough with five oil pomaces from black seed, pumpkin, hemp, milk thistle and primrose on the gluten structure was studied by Rumińska et al. [92] with application of FT-IR and FT-Raman spectroscopy. Analysis of the spectroscopic results indicated that the observed changes depended mainly on the type and amount of fatty acids present in the pomaces although pomaces contained considerable amount of dietary fibre. If the pomaces contain a low number of fatty acids, aggregated β-sheets with intermolecular hydrogen bonds were formed from β-turns and antiparallel-β-sheets. Whereas non-aggregated β-structures were observed for pomaces with a high number of fatty acids. Reduction in the quality of bread supplemented with DFP can be caused by dehydration of the gluten network during dough mixing [93,94]. Therefore, application of preliminary moisturizing of DFP can eliminate negative changes in the gluten structure. Nawrocka et al. [95] used FT-IR and FT-Raman techniques to determine changes in the gluten structure after addition of eight DFP (in concentrations 3%, 6% and 9%) which had been moistened for 30 min before dough mixing. Modification of the gluten proteins with pre-moistened apple, flax and oat fibres did not cause significant changes in the secondary structure, whereas addition of the rest of moisturized preparations resulted in appearance of the band at 1596 cm^−1^, which can be assigned to hydrated/extended β-sheets. These structures can be formed by parallel-β-sheets and/or H-bonded β-turns which suggests that structural changes concerned mainly glutenins. Analysis of disulphide bridges region indicated that DFP moisturizing affected conformation of these bonds. However, any significant changes were not determined for water populations and aromatic amino acids microenvironment. Based on these results it can be claimed that pre-moisturizing of some fibre preparations (chokeberry, carob, cranberry and cacao) can prevent aggregation of the gluten network.

Some studies concern interactions between gluten and DFP without starch (in the gluten dough). Addition of seven DFP to the gluten dough induced similar changes in the secondary structure of gluten proteins regardless of their botanical origin and chemical composition [96]. The most noticeable changes were observed in the α-helix and antiparallel β-sheets region. The results suggested that a fibre component which was present in all DFP led to formation of aggregated structures in the form of antiparallel β-sheets from α-helices from two protein complexes. Changes in the protein structure could be attributed to the cellulose because of its highest content in the analysed fibres. Other structural changes concerned mainly β-sheets, β-turns, disulphide bridges conformations and TYR and TRP residues were probably associated with other compounds present in the DFP. Changes in secondary structure and water populations in the gluten dough supplemented with wheat bran were studied by Bock and Damodaran [36]. Addition of wheat bran to the gluten dough resulted in redistribution of water between its free and bound state in the gluten matrix. In the case of secondary structure, wheat bran caused decrease in β-turns content, increase in the number of β-sheets and random structures whereas the α-helix content remained unchanged regardless of dough moisture content. The results suggested that after bran addition the preferred structures of gluten protein in hydrated state are β-sheets and random coils which may be a result of water redistribution and partial dehydration of gluten proteins by wheat bran. The transformation of β-turns into β-sheets can be responsible for reduction of bread quality.

Changes in the gluten structure can be also studied after baking. Sivam et al. [97] determined structural changes of the gluten network in model bread supplemented with pectin and blackcurrant polyphenols. Addition of blackcurrant polyphenol extract and pectin caused a reduction in the number of β-turns and appearance of the band associated with unordered structures. Additionally, the smaller amount of α-helix and slightly higher amount of intermolecular H bonds was observed compared to the control sample. It was concluded that polyphenols and pectin compete with flour protein for water molecules and can be enclosed in the hydrophobic pocket formed by gluten side chains. It resulted in the entanglement of the side chains of glutamine between neighbouring molecules as well as different parts of the same molecules.

#### 4.5.2. Polysaccharides

Polysaccharides are the most widely used ingredients in the food industry. They can act as thickeners, gelling agents, antistalling agents, emulsifiers, stabilizers, fat replacers, and can have applications in the fields of edible films, flavour encapsulation, and inhibition of crystallization [98,99,100,101]. In addition, they have a great impact on dough rheology and bread quality and thus structure of gluten network [1]. The effects of polysaccharides on the functional properties and quality of the wheat dough and bread depend on the polysaccharide structure, origin, particle size, and the dosages of the polysaccharide incorporated into the dough [101]. Among the polysaccharides cellulose and its derivatives, pectins, dextrins, gums, arabinoxylans etc. are used the most often.

Correa et al. [56] studied interactions between hydrocolloids (microcrystalline cellulose (MCC), carboxymethylcellulose (CMC), hydroxypropylmethylcelluloses (HPMC), low methoxylated (LMP) and a highly methoxylated citrus pectin (HMP)) and the gluten network in the wheat dough. Modified celluloses and pectins were used at 1.5% and 2.0% flour base, respectively. In general, all hydrocolloids increased content of parallel β-sheet and antiparallel-β-sheets. There were no major changes in content of β-turns in dough supplemented with HPMC, LMP and HMP. β-turn content increased considerably in the CMC-gluten samples, whereas MCC addition led to lower its content. The biggest increase in α-helix conformation was observed in dough supplemented with HMP followed by HPMC, LMP and CMC. Whereas the lowest α-helix content and an increase in the amount of unfolded conformations can be seen in CMC dough. Effect of the carboxymethylcellulose (CMC) in two concentrations (0.5% and 1%) on the gluten structure in the wheat dough was also studied by Zhao et al. [102]. Generally, an increase in the amount of intramolecular β-sheets and antiparallel β-sheets contents at the expense of α-helix content was observed comparing to control gluten. Moreover, addition of 0.5% CMC led to great reduction in α-helix content. Contents of β-turns and β-sheets remained unchanged. However, addition of 1% CMC resulted in increase of α-helices, β-sheets and intramolecular β-sheets at the expense of β-turns. Interactions between gluten and four hydrocolloids (xantham gum (XG), guar gum (GG), locust bean gum (LBG) and highly methoxylated pectin (HMP)) in wheat dough were studied by Linlaud et al. [47] with application of FT-Raman spectroscopy. Among the studied polysaccharides, only LBG caused opposite changes in the secondary structure of gluten comparing to the other polysaccharides (XG, GG and HMP). LBG addition decreased contents of β-sheets, β-turns and antiparallel-β-sheets with simultaneous increase in α-helix content. Considerable changes in the disulphide bridges conformation after supplementation were observed including reduction in the amount of S=S bonds in the g-g-g conformation, appearance of the S=S bonds in the t-g-g conformation, and increase in the t-g-t conformation. Additionally, some changes were determined for TYR and TRP residues. FT-IR spectroscopy also was used to determine effect of water-soluble resistant dextrin on the gluten secondary structure in wheat dough [103]. Analysis of the spectra showed increase in the β-sheets at the expense of β-turns with increasing dextrin concentration. Contents of α-helices and random coils remained constant. The results indicated aggregation of gluten proteins by the transition of β-turns into β-sheets.

Interactions between gluten proteins and polysaccharides can be studied in two types of dough—model dough and gluten dough (without starch). Both doughs supplemented with dietary fibre polysaccharides (microcrystalline cellulose, inulin, apple and citrus pectins) in five concentrations (3%, 6%, 9%, 12% and 18%) were used in the studies of Nawrocka et al. [57,104,105]. Structural changes were determined with application of FT-Raman and FT-IR spectroscopies. Analysis of the spectroscopic results concerning both dough types showed that the studied polysaccharides induced formation of aggregated β-structures e.g., antiparallel-β-sheets with intermolecular hydrogen bonds, hydrogen bonded β-turns. Presence of these structures suggests dehydration of the gluten network as a result of competition for water between gluten and polysaccharides during dough mixing. Differences in the amount and type of the aggregated structures observed as a result of dough supplementation with polysaccharides can be associated with solubility of the polysaccharides in water. Cellulose and inulin as water insoluble polysaccharides led to formation of pseudo-β-sheets, whereas water soluble pectins induced formation of hydrated/extended β-sheets. Differences in the gluten structure between gluten dough and model dough were observed by Nawrocka et al. [33] as a result of supplementation of both doughs with celluloses differing in particle size and chemical structure. Analysis of the FT-IR spectra of the control gluten dough showed presence of the bands assigned to hydrated β-sheets and water microvoids which indicate proper hydration level of the gluten network. However, model dough—celluloses spectra showed structures (H bonded β-turns and antiparallel-β-sheets) characteristic for gluten network of low hydration. Additionally, analysis of amide III band confirmed incorporation of celluloses into gluten network regardless of the dough type. Effect of other polysaccharide, konjac glucomannan (KGM), on the gluten structure in both doughs was also studied. Zhou et al. [106] supplemented wheat dough (at two water absorptions −55% and 81%) with KGM in two concentrations (2.5% and 5%). A decrease in α-helix content with simultaneous increase in β-sheets and β-turns content was determined after KGM addition. However, increasing water absorption had greater influence on conformation of disulphide bridges and TYR residues than on the secondary structure, leading to increase in the tyrosine doublet value and amount of the S=S bonds in the g-g-g conformation. Structural changes in the gluten dough supplemented with KGM were also studied by Wang et al. [107] and Li et al. [108] with application of FT-IR spectroscopy. In research carried out by Li et al. [108], α-helix and β-sheets contents increased, while contents of β-turns and random coils decreased with increasing concentration of KGM. Opposite results concerning secondary structure were obtained by Wang et al. [107]. Another polysaccharide used in the wheat dough supplementation was water extractable arabinoxylan (WEAX) [109,110,111]. Zhu et al. [110] studied changes in conformation of gluten proteins caused by 5% addition of high and low molecular weight WEAX (HMW and LMW WEAX). FT-IR analysis revealed an increase in β-sheets content and a decrease in α-helical structures. Analysis of the percentage distribution of disulphide bridge conformations showed that the number of S=S bonds in the t-g-g conformation decreased, while the number of S=S bonds in the t-g-t conformation remained unchanged, especially after addition of HMW WAEX. Additionally, WEAX induced burriedness of TRP and TYR residues within protein complex. Similar results were presented in other studies of the same Authors [109]. Lower concentrations of WEAX (1% and 2%) caused opposite changes in the gluten structure [111]. With the incorporation of WEAX, no considerable changes were observed in the amide I band. The only noticeable changes were a small decrease in α-helices and increase in β-turns. On the contrary, an increase in the g-g-g and t-g-g conformations at the expense of t-g-t conformation of disulphide bridges and exposition of the TRP residues at the protein surface were observed. Other studied polysaccharide was dextran [112]. 5% addition of dextran to gluten dough resulted in a decrease in antiparallel β-sheets and β-sheets contents, and an increase in α-helix and β-turn contents. Addition of dextran in slight acidic environment (pH = 5.5) resulted in an increase of antiparallel β-sheets content comparing to dextran-gluten sample but in a decrease comparing to control sample.

Different polysaccharides were also added to the gliadins. Secundo and Guerrieri [66] studied interactions between gliadins and dextrin at 50% concentration. Dextrin addition caused a decrease in the content of β-sheets with intramolecular hydrogen bonds. It may indicate changes in the pattern of hydrogen bonds formation during dough mixing in the presence of dextrin. Chourpa et al. [113] studied effect of arabic gum addition on the gliadin structure under different pH conditions. Analysis of Raman spectra indicated that the strongest interactions between gliadins and arabic gum occurred at pH 3.0. additionally, the polysaccharide did not affect α-helical structures at pH 3.0–4.0, while decrease in α-helices content was observed at pH lower than 2.5. Lowering pH below 3.0 led to an increase in β-sheets, whereas at pH > 3 their content was constant. The bands assigned to β-turns and random coils appeared unchanged in the presence of arabic gum at all pH values. The amide III region and TYR bands of arabic gum-protein complexes also did not show any noticeable changes.

#### 4.5.3. Polyphenols

Polyphenols such as phenolic acids, anthocyanins, flavonoids or tannis can be added to wheat dough in the free form as well as polyphenols extracts or in dietary fibre preparations [63]. Monomeric polyphenols (e.g., phenolic acids) as antioxidants can decrease gluten strength by reduction of disulphide bridges leading to formation of weak and less elastic gluten network. On the contrary, high molecular weight polyphenols (e.g., tannins) may strengthen the gluten network by formation of cross-links through hydrogen and hydrophobic interactions [114,115].

Monomeric polyphenols like five phenolic acids (cinnamic, p-coumaric, caffeic, ferulic and chlorogenic acids) in concentrations 0.05%, 0.1% and 0.2% acids were added to the model dough in the studies of Krekora et al. [63]. Analysis of the amide I band indicated the development of aggregated structures such as pseudo-β-sheets and hydrogen bonded β-turns in modified gluten network. As a result of the addition of caffeic and chlorogenic acids, characterized by the highest antioxidant activity, the biggest changes in the conformation of disulphide bridges and tryptophan environment were observed. However, tyrosine environment was the least influenced by the phenolic acids. These studied showed that the structural changes observed in gluten proteins can depend on the antioxidative properties of the phenolic acids. Effect of ferulic acids, which is the most abundant in wheat, on the gluten structure in whole wheat model system were studied by Huang et al. [91]. Increase in the ferulic acid concentration from 1% to 4% caused increase in the α-helix and β-turn contents with simultaneous decrease in β-sheets and random coils. The Authors claimed that reduction in the β-sheets at the expense of β-turns can lead to damage of the gluten network.

Flavonoids including anthocyanins can be also regarded as monomeric polyphenols. Effect of selected anthocyanins (malvin, pelargonin, oenin, cyanin, kuromanin, callistephin) on the gliadin structure was studied by Taddei et al. [116] with using of FT-Raman spectroscopy. Malvin and pelargonin caused a similar rearrangement involving a substantial increase in β-sheet and a simultaneous decline in β-turn and unordered structures. No changes in the amide I band were detected after treatment with oenin, kuromanin, callistephin and cyanin. All anthocyanins led to an increase in the t-g-g conformation at the expense of g-g-g conformation of disulphide bridges. In the case of tyrosine doublet, only malvin caused significant decrease in the ratio value indicating burriedness of the TYR residues in the protein complex. The spectral results showed that only malvin interacted with gliadins leading to considerable changes in their structure. Opposite changes in the gliadin secondary structure were observed after application of cyanidin and 3-ethoxycarbonylcoumarin [55]. Mazzaracchio et al. [117] used FT-IR spectroscopy to study structural changes in the gliadins modified by compounds similar to those of Taddei et al. [116]. Addition of keracynin and malvin caused an increase in β-sheets with simultaneous decrease in random coils. Slight changes were reported for α-helices and β-turns. FT-IR and FT-Raman spectroscopies also were used to study gliadin structure upon treatment with quercetin at various values of pH [118]. Quercetin addition at pH 5 led to increase in the α-helix and β-sheet contents at the expense of β-turns and random coils. Simultaneously, disulphide bridges transformed from g-g-g into t-g-g and t-g-t conformations. At pH < 5.0, TRP and TYR residues were exposed at the protein complex as a result of quercetin addition, whereas increase in the pH above 7.0 led to burriedness of the amino acid residues inside protein complex. However, FT-IR spectroscopy has proved to be ineffective to study the effect of sorghum and grape proanthocyanidins supplementation on the gluten structure in the studies of Girard et al. [119] due to overlapping of the amides bands with proanthocyanidins bands.

Effects of tannins as polymeric polyphenols on the gluten structure were also studied. Wang et al. [120] supplemented wheat dough with synthetic tannins. It was observed an increase in β-turns and α-helices with a decrease in β-sheets. The addition of tannins induced aggregation of gluten proteins resulting in changes of the gluten microstructure and thus improvement of dough mixing properties. Opposite structural changes were observed after addition of high concentration of persimmon tannin (8%) to the wheat dough by Du et al. [121]. Addition of the polyphenols caused different rearrangements within structure of gluten proteins. It appears that type of used polyphenol, its concentration as well as pH value can induce different changes in the gluten structure. It seems to be difficult to determine particular changes in gluten structure assigned to specific polyphenol because different factors are used to change gluten structure.

### 4.6. Other Factors

Due to its unique functional properties, wheat gluten can also be used as stabilizing agent, gelling agent, thermoplastic material etc. [122]. Heat-induced changes in the gluten structure during gel formation was studied by Wang et al. [123] with application of FT-IR spectroscopy. FT-IR spectra indicated that a low temperature of gelation (40–50 °C) caused appearance of random coils with simultaneous decrease in α-helix content. It suggested protein unfolding and rearrangement of gluten proteins. Increase in the gelation temperature to 60 °C led to disappearance of random coil and formation of β-turns. Further increase in temperature to 90 °C was related to protein aggregation observed as increase in the amount of intermolecular β-sheets. Gluten proteins, especially gliadins, can be also used to form biodegradable films, which properties are similar to polymer films [124]. Mangavel et al. [65] studied structural changes in gliadins during film formation by drying. Analysis of the amide I band during drying at 70 °C indicated considerable increase in the content of hydrogen bonded β-sheets, whereas α-helix and random coil contents were not affected. These studies also showed dependence between drying conditions (temperature and time of drying) and types of secondary structures formed. Drying in high temperatures during short time favoured hydrogen bonded β-sheets, α-helices and probably random coils at the expense of other kinds of β-sheets and β-turns.

Wheat gluten applications in food industry are limited by its low solubility. The gluten solubility can be improved by deamidation of uncharged amino acids e.g., glutamine and asparagine and hydrolysis among others. Deamidation is the transformation of amides into carboxyl groups. Liao et al. [17] used succinic and citric acids to deamidate gluten amino acids and determined changes in its structure by using FT-IR and Raman spectroscopy. Deamidation by citric acid led to greater increase in the intermolecular and extended β-sheets content at the expense of α-helices and β-turns comparing to succinic acid. Additionally, analysis of the Raman peaks assigned to SH groups showed increase in the amount of SH groups after deamidation by both acids. The results indicated that deamidation affected hydrogen bonding pattern rather than peptide bonds. Gluten solubility was also changed by partial hydrolysis using alcalase and microbial transglutaminase [125]. The partial hydrolysis by both enzymes led to conversion of α-helices and β-turns into β-sheets and random coils, which indicated depolymerisation of the gluten peptide chains. However, excessive hydrolysis caused protein aggregation.

## 5. Application of Spectroscopic Methods in Wheat Products Technology

Spectroscopic methods also can be used to determine parameters crucial from the technological point of view. In this case, both NIR and MIR infrared spectroscopies and Raman spectroscopy can be used. Sinelli et al. [126] used FT-NIR spectroscopy to predict technological quality of semolina from whole durum wheat kernels (without milling or grinding). The Authors tried to correlate the spectral data with protein content, gluten content, gluten index and alveographic indices by development of PLS calibration models. Only protein content was well correlated with the spectral data. Attempts to determine gluten content in wheat flour were taken by Czaja et al. [127], who applied four spectroscopic method: FT-Raman, NIR, ATR-IR and DRIFTS (diffuse reflectance infrared Fourier transform spectroscopy) spectroscopies. The study confirmed possibility of using these four methods combined with multivariate calibration to analyse gluten content in wheat flour. The standard error of prediction ranged from 3% to 6%. The protein and ash contents in commercial wheat flour were determined successfully by using HATR (horizontal attenuated total reflectance) mid-infrared spectroscopy combined with partial least squares (PLS) regression [128]. A spectral region 1300–1700 cm^−1^ were assigned to protein content, whereas a band connected with ash content was very difficult to find. There were also trials to use Raman spectroscopy to determine ash and moisture in a wheat flour [129]. Analysis of Raman spectra and reference data combined with partial least squares (PLS) regression models allowed for fast and reliable determination of these two parameters with standard error of prediction was ca. 2%. Confocal Raman microspectroscopy were used to investigate structure of a wheat grain on a microscopic scale [130]. The studies were focused on the determination of protein content, composition of the starchy endosperm, and content of arabinoxylans and ferulic acid derivatives in the alleurone cells. Additionally, the protein content and changes in the secondary structure of the proteins, especially in α-helix content, were proved to be connected with grain hardness.

## 6. Conclusions

The present review proves that infrared and Ramam spectroscopies are adequate tools to determine structure of gluten network and gluten proteins. Each of methods provides valuable information about changes in the secondary structure but to get more complete understanding of the gluten structure, both methods should be used. Additionally, gluten in different state (in hydrated state, powder or in solution) can be studied with application of both methods. Analysis of the studies presented in the review showed that water is a main factor affecting gluten structure. Gluten network of low hydration level forms other kind of secondary structures than gluten network of high hydration level i.e., dry gluten contains mainly unordered structures, whereas α-helices, β-sheets and β-turns were formed with increasing hydration level. These results are in agreement with the ‘loops and trains’ model of gluten network proposed by Belton. Water is also important in the dough supplementation with different compounds. Depending on the supplement used, competition for water between gluten proteins and supplement may take place. For example, dietary fibre preparations and polysaccharides caused dehydration of gluten network during mixing, whereas this phenomenon is not observed after polyphenols addition. Dehydration leads to formation of aggregated β-structures with intra- and intermolecular hydrogen bonds. Opposite effect on the gluten secondary structure was observed after emulsifiers addition, which induced formation of α-helices from β-sheets and β-turns. The presented results indicated that structural changes, induced by physical and chemical factors, concerned mainly β-structures. β-structures, especially β-sheets, can be considered as structures responsible for the formation of gluten network characterized by proper structure and mechanical properties.

## Figures and Tables

**Table 2 molecules-26-00508-t002:** Assignments of particular secondary structures in the amide I band for FT-IR and Raman spectroscopy.

Secondary Structure	FT-IR Spectroscopy (cm^−1^)	Raman Spectroscopy (cm^−1^)
**Non-Aggregated Secondary Structures**		
α-helix	1650–1649 [19,50,51] 1655 [34,52,53,54]	1660–1650 [29,55,56,57] 1670 [55]
β-sheet	1687–1680 [50,51,54,58]	1633–1619 [29,47,57,59]
antiparallel-β-sheet	1632–1629 [50,60]	1695–1675 [29,33,47,56,57]
β-turns	1677–1666 [34,50,51,54,61]	1677–1666 [29,47,56]
**Aggregated secondary structures**		
pseudo-β-sheets	1615–1613 [50,54,62]	1625–1610 [29,57,63]
α-helix H-bonded with water	1651–1648 [51,53,61,64]	-
hydrated β-sheets	1630 [51,54] 1623–1600 [65]	1607–1606 [63]
H-bonded antiparallel-β-sheets	1690 [66] 1627 [66]	1680 [57]
H-bonded β-sheets	1695–1680 [66]	1682 [63]
H-bonded β-turns	1643 [66]	1646 [57] 1656 [63]

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
