# Peer review of "Effects of Physical and Chemical Factors on the Structure of Gluten, Gliadins and Glutenins as Studied with Spectroscopic Methods"

_molecules, 2021, doi:10.3390/molecules26020508_

Round 1
Reviewer 1 Report
The review "Spectroscopic methods as analytical tools to determine changes in the structure of gluten, gliadins and glutenins" describes the investigation of the structure of the dough proteins using spectroscopic methods. The tremendous significance of dough making and the widespread availability of spectroscopic techniques, namely FTIR and Raman spectroscopies, make the topic of this review highly important.
The review covers a wide range of research dealing with the structural changes of dough proteins and the effect of hydration, additives, heat, and other factors on their structure. It thus can provide a good starting point for the researcher interested in this field.
That said, in my opinion, in its current form, the manuscript falls short of this target if that was indeed the idea behind the review.
If I were to enter this field, my main questions would be "what was done, and how it was done." The review outlines the former but speaks very little of the latter. The experimental techniques' description is minimal, and no attention is paid to the sample preparation at all. A subject of the comparison of IR/Raman and how to choose one or the other is explained very briefly. Based on the title of the article, I would expect more attention paid to such subjects.
On the other hand, if the authors intended to critically review the information of the structure of the dough proteins and the effects having the impact on that available so far, then it seems that the manuscript lacks a narrative that could somehow help to get a sense of the overall trends in this field of research. Some of it is provided in the conclusions but again very briefly. And if this was the aim of the authors, then the title is misleading.
That said, the majority of the work is already done, and the authors need to provide some focus to the review.
Some minor details also need polishing. Below are some things I noticed (not an exhaustive list):
Table 1 Formatting – the same references are given for each line of the table, probably can only be mentioned once for the whole line
144 "spectroscopy were used"
246 "can induced"
248 MIR – while it is understandable from the context, it would be better to have a list of abbreviations at the start of the article
263, 265 "water absorption" – not clear if it means light absorption of hydration
Chapter 4.2 – specific methods are not mentioned at all
308 HMW GS – same as with MIR above
338 "known to worsened"
430 "competition" must be "compete"
448-449 Similar… similar – two sentences in succession starting with "similar".
548-550 "It was observed…" – sentence must be rewritten
679 FTIR inadequate – why?
FT-IR/FTIR – inconsistent spelling throughout the article.
709-710 “hydrogen bonded β-sheets (…) at the expense of other kinds of β-sheets” – are there non-hydrogen bonded β-sheets?
730 "spectral data were tried to connect" – must be rewritten
767 "Dehydratation"
Authors must improve the style of the text. Currently, some of the descriptions are repetitive and lengthy, and overall the article is hard to read, e.g., the paragraph from 421 to 480 is almost a page long!
Overall, I feel the review has the potential to be a nice and informative one, but authors must spend some time improving it to realize its potential.
Author Response
ANSWERS TO THE COMMENTS OF REVIEWER 1
We would like to thank the Reviewer for valuable comments which have helped us to improve quality of our work. The answers to the Reviewer’s specific comments are listed below:
- The review covers a wide range of research dealing with the structural changes of dough proteins and the effect of hydration, additives, heat, and other factors on their structure. It thus can provide a good starting point for the researcher interested in this field. That said, in my opinion, in its current form, the manuscript falls short of this target if that was indeed the idea behind the review. If I were to enter this field, my main questions would be "what was done, and how it was done." The review outlines the former but speaks very little of the latter. The experimental techniques' description is minimal, and no attention is paid to the sample preparation at all. A subject of the comparison of IR/Raman and how to choose one or the other is explained very briefly. Based on the title of the article, I would expect more attention paid to such subjects. On the other hand, if the authors intended to critically review the information of the structure of the dough proteins and the effects having the impact on that available so far, then it seems that the manuscript lacks a narrative that could somehow help to get a sense of the overall trends in this field of research. Some of it is provided in the conclusions but again very briefly. And if this was the aim of the authors, then the title is misleading.
We thank the Reviewer for the comment. We agree with the Reviewer that the title of the article can be misleading. For this reason, the title has been changed to “Effects of physical and chemical factors on the structure of gluten, gliadins and glutenins studied with application of spectroscopic methods”. In this article, we want to show that spectroscopic methods can be applied to study structure and structural changes in the gluten network and gluten proteins (gliadins and glutenis). Usually, spectroscopic methods, especially FT-IR, are used to study secondary structure of the proteins in aqueous solution. However, none of the proteins discussed in the article are water-soluble. Additionally, gluten network is a very complex matrix.
When preparing the article, we tried to ensure that the number of words in the article did not exceed 10 000. Commonly, journals ask the Authors do not exceed this word limit. For this reason, we decided not to include details on the samples preparation in this article, because we could exceed significantly the word limit. Additionally, the references contain the detailed descriptions of the sample preparation and a reader of the article can get it from the original paper.
As for the comparison of the spectroscopic methods, we did not want to compare the FT-IR and FT-Raman spectroscopy because these methods are regarded as complimentary. Our aim was to present what kind of information we can get from each method. A reader/experimenter should decide itself which method he wants to use in his studies depending what kind of information he wants to get. A reader/experimenter can find these information in Chapter 3.
- Table 1 Formatting – the same references are given for each line of the table, probably can only be mentioned once for the whole line
The reference numbers were placed in the first column of the Table 1.
- 144 "spectroscopy were used"
The verb ‘were’ has been changed to ‘was’ (L.154).
- 246 "can induced"
It has been changed to ‘can induce’ (L.262).
- 248 MIR – while it is understandable from the context, it would be better to have a list of abbreviations at the start of the article
List of abbreviations has been added to the manuscript after Conclusions (L.817-829)
- 263, 265 "water absorption" – not clear if it means light absorption of hydration
We thank the reviewer for the comment. The term ‘water absorption’ is connected with amount of water added to the dough at the beginning of the dough mixing and is commonly used in the studies concerning wheat dough. The parameter ‘water absorption’ can be determine with application of farinograph according to the standard method ICC 115/1. The amount of water added to the dough is a crucial parameter for the formation of dough characterized by proper consistency. The presented studies concerned doughs with different water absorptions and consequently different consistency. Dough consistency was higher at 35% water absorption than 90% water absorption.
- Chapter 4.2 – specific methods are not mentioned at all
The specific methods were added to the chapter 4.2. (L.320-325): “ATR-FT-IR studies of van Velzen et al. [78] showed that dough stretching resulted in in-crease in the extended β-sheets with simultaneous considerable decrease in α-helix content. Slightly different results obtained Wellner et al. [79], who studied changes in gluten structure during a few cycles of extension and relaxation of gluten isolated from developing grain with application of dynamic FT-IR.”
- 308 HMW GS – same as with MIR above
List of abbreviations has been added to the manuscript after Conclusions (L.817-829).
- 338 "known to worsened"
It has been changed to ‘known to worsen’ (L.357).
- 430 "competition" must be "compete"
The sentence containing this word has been deleted (L.454).
- 448-449 Similar… similar – two sentences in succession starting with "similar".
The second sentence of the two indicated has been rewritten (L.472-476): “The analysis of TYR and TRP behaviour also indicated gluten proteins aggregation. Additionally, Nawrocka et al. [29] obtained similar results in the previous research concerning interaction between DFP in concentration 6% and gluten proteins in the model flour.”
- 548-550 "It was observed…" – sentence must be rewritten
The sentence has been rewritten (L.582-584): “Generally, an increase in the amount of intramolecular β-sheets and antiparallel β-sheets contents at the expense of α-helix content was observed comparing to control gluten.”
- 679 FTIR inadequate – why?
We thank the Reviewer for the comment. The word ‘inadequate’ should be change to ‘ineffective’. The sentence has been rewritten (L.718-721): “However, FT-IR spectroscopy has proved to be ineffective to study the effect of sorghum and grape proanthocyanidins supplementation on the gluten structure in the studies of Girard et al. [119] due to overlapping of the amides bands with proanthocyanidins bands.”
- FT-IR/FTIR – inconsistent spelling throughout the article.
We adopted the spelling of the method as FT-IR throughout the manuscript.
- 709-710 “hydrogen bonded β-sheets (…) at the expense of other kinds of β-sheets” – are there non-hydrogen bonded β-sheets?
We thank the Reviewer for the comment. The β-sheets regarded as basic secondary structure contain hydrogen bonds and they are called parallel or antiparallel β-sheets without using the term ‘hydrogen bonded’. The term ‘hydrogen bonded β-sheets’ is commonly used to describe aggregated β-sheets. These hydrogen bonds can be formed between parallel or antiparallel β-sheets, α-helices etc. They can be observed in two forms as intra- and interchain hydrogen bonds. As it was mentioned in the Chapter 2 (Biochemistry and structure of gluten network and gluten proteins), intra- and interchain hydrogen bonds are important for formation of gluten network characterized by proper structure and mechanical properties.
- 730 "spectral data were tried to connect" – must be rewritten
The sentence has been rewritten (L.771-773): “The Authors tried to correlate the spectral data with protein content, gluten content, gluten index and alveographic indices by development of PLS calibration models.”
- 767 "Dehydratation"
It is a spelling mistake. There should be ‘dehydration’ instead of ‘dehydratation’. It was corrected in the text (L. 807).
- Authors must improve the style of the text. Currently, some of the descriptions are repetitive and lengthy, and overall the article is hard to read, e.g., the paragraph from 421 to 480 is almost a page long!
We thank Reviewer for the comment. We improve the style of the manuscript. Some minor corrections are in the text and they are not mentioned in the list of improvements. Major improvements are listed below:
- 63-64 – “As for the secondary structure, α-/β- and γ-gliadins contain mainly α-helices and β-sheets, whereas β-turns dominate in ω-gliadins [6].”
- 67-69 – “The glutenin subunits (GS) are characterized by alcohol solubility similar to gliadins after treating them with reducing agent caused cleavage of disulphide bridges.”
- 86-88 – “As it was mentioned above, gliadins form only intrachain disulphide bridges, whereas glutenins participate in intra- and interchain S=S bonds.”
- 100-103 – “Spectroscopic methods can be used to study secondary structure of native and commercially available gluten. Additionally, the gluten samples can be studied in different states – powder and hydrated gluten dough. Also different spectroscopic methods can be applied depending on what kind of structural information a scientist wants to obtain.”
- 138-139 – “Similar behaviour was observed in the peptides mimicking the HMW subunits by Feeney et al. [25].”
- 142-145– “An increase in the hydration level led to an increase in the β-turn content at the expense of β-sheets for the short peptides, whereas formation of intermolecular β-sheets was observed for the long peptides.”
- 152-153 – “The FT-IR spectroscopy has been used the most often to determine secondary structure of the gluten proteins.”
- 169-172 – “To study structure of the gluten proteins also Fourier transform Raman spectroscopy (FT-Raman) is used. Raman spectrophotometer is equipped in infrared Nd:YAG laser with λ = 1064 nm.”
- 188-202 – “Amide II band can be also used to determine secondary structure of the proteins. However, it is less reliable for this purpose comparing to amide I and amide III bands. It can be applied to determine changes in the proteins hydration because increase in the dough hydration level causes shift of the amide II band toward higher wavenumbers [34,35]. It is a well-known fact that water is necessary to obtain wheat dough characterized by appropriate mechanical properties. Water populations present in the gluten network can be determined by the analysis of the spectral region assigned to OH stretching (2800 – 4000 cm-1) in the FT-IR spectra. Water molecules interact with gluten polypeptide chains by strong and weak hydrogen bonds with characteristic bands at ca. 3055 and 3190 cm-1, respectively [36]. Additionally, other bands in the OH stretching region were assigned to water molecules participating in two hydrogen bonds (ca. 3280 cm-1) [37], small hydrogen-bonded water clusters (ca. 3370) [36] and free water (ca. 3650 cm-1) [37].”
- 248-251 – “Chemical changes in the gluten network during dough mixing have been studied with application of Fourier transform infrared spectroscopy (FT-IR) with analysis of amide III band. This band was chosen to study secondary structure of hydrated dough due to lack of water oscillations.”
- 264-266 – “Attempts were also made to assign the amide bands from MIR spectra to bands from NIR spectra. Both kinds of spectra were registered during dough mixing [71].”
- 293-294 – “However, decrease in the temperature induced formation of different structural changes in the studied doughs.”
- 375-384 – “Detailed analysis of the amide I band showed increase in the α-helix content that were accompanied by a decrease in the amount of β-sheets. FT-Raman spectra provide also information about disulphide bridges. Analysis of the spectral region connected with S=S bonds show that only 0.25% addition of SSL resulted in the appearance of S=S bonds in the t-g-t conformation. Additionally, changes in the microenvironment of TYR and TRP were determined. Tyrosine doublet ratio increased in the sample containing 1% of SSL, whereas decreased for samples with 0.5% and 0.25% SSL content. It indicated exposition at the protein complex surface and burriedness in the protein complex of the TYR residues, respectively.”
- 443-478 – “A common wheat flour contains native fibre substances that may interfere interactions between gluten network and DFP. Therefore, a model flour, reconstituted from wheat starch and wheat gluten in a constant weight ratio (80:15 w/w), is often used in this kind of studies, nowadays. Structural changes in the gluten network obtained from model dough can be studied with application of FT-Raman and FT-IR spectroscopies. Gluten samples from model dough supplemented with seven DFP in the concentration of 3 – 18% were investigated by Nawrocka et al. [88] with application of FT-Raman spectroscopy. Analysis of the difference spectra in the amide I band showed that all dietary fibres except oat induced similar changes in the secondary structure of gluten proteins. The most noticeable differences were detected in the regions associated with hydrogen bonded β-sheets and β-turns. The DFP also affected the conformations of disulphide bridges. The rise in the fibre concentration resulted in an increase in the number of S=S bonds in the stable g-g-g conformation in the case of cacao and decrease for the rest of fibres except for oat. The results indicated that increasing content of DFP in the case of carrot and carob induced the cleavage of the intrachain disulphide bonds. The opposite effect, the appearance of S=S intrachain bonds, was observed after addition of the higher concentration of cranberry and flax preparations. Conversion of the disulphide bridges from g-g-g conformation into less stable t-g-g and t-g-t conformations could result in the gluten proteins aggregation. The analysis of TYR and TRP behaviour also indicated gluten proteins aggregation. Additionally, Nawrocka et al. [29] obtained similar results in the research concerning interaction between DFP in 6% concentration and gluten proteins in the model flour. These studies indicated that the aggregation of gluten proteins can be the result of competition for water between gluten and DFP during dough mixing [89,90].”
- 531-538 – “The results suggested that a fibre component which was present in all DFP led to formation of aggregated structures in the form of antiparallel β-sheets from α-helices from two protein complexes. Changes in the protein structure could be attributed to the cellulose because of its highest content in the analysed fibres. Other structural changes concerned mainly β-sheets, β-turns, disulphide bridges conformations and TYR and TRP residues were probably associated with other compounds present in the DFP.”
- 549-551 – “Changes in the gluten structure can be also studied after baking. Sivam et al. [97] determined structural changes of the gluten network in model bread supplemented with pectin and blackcurrant polyphenols.”
- 613-615 – “Differences in the amount and type of the aggregated structures observed as a result of dough supplementation with polysaccharides can be associated with solubility of the polysaccharides in water.”
- 638-639 – “Another polysaccharide used in the wheat dough supplementation was water extractable arabinoxylan (WEAX) [109–111].”
- 688-690 – “These studied showed that the structural changes observed in gluten proteins can depend on the antioxidative properties of the phenolic acids.”
- 705-706 – “The spectral results showed that only malvin interacted with gliadins leading to considerable changes in their structure.”
- 812-815 – “The presented results indicated that structural changes, induced by physical and chemical factors, concerned mainly β-structures. β-structures, especially β-sheets, can be considered as structures responsible for the formation of gluten network characterized by proper structure and mechanical properties.”
The paragraph in L.421-480 was divided into 3 paragraphs (L. 443-510):
“A common wheat flour contains native fibre substances that may interfere interactions between gluten network and DFP. Therefore, a model flour, reconstituted from wheat starch and wheat gluten in a constant weight ratio (80:15 w/w), is often used in this kind of studies, nowadays. Structural changes in the gluten network obtained from model dough can be studied with application of FT-Raman and FT-IR spectroscopies. Gluten samples from model dough supplemented with seven DFP in the concentration of 3 – 18% were investigated by Nawrocka et al. [88] with application of FT-Raman spectroscopy. Analysis of the difference spectra in the amide I band showed that all dietary fibres except oat induced similar changes in the secondary structure of gluten proteins. The most noticeable differences were detected in the regions associated with hydrogen bonded β-sheets and β-turns. The DFP also affected the conformations of disulphide bridges. The rise in the fibre concentration resulted in an increase in the number of S=S bonds in the stable g-g-g conformation in the case of cacao and decrease for the rest of fibres except for oat. The results indicated that increasing content of DFP in the case of carrot and carob induced the cleavage of the intrachain disulphide bonds. The opposite effect, the appearance of S=S intrachain bonds, was observed after addition of the higher concentration of cranberry and flax preparations. Conversion of the disulphide bridges from g-g-g conformation into less stable t-g-g and t-g-t conformations could result in the gluten proteins aggregation.. The analysis of TYR and TRP behaviour also indicated gluten proteins aggregation. Additionally, Nawrocka et al. [29] obtained similar results in the previous research concerning interaction between DFP in concentration 6% and gluten proteins in the model flour. These studies indicated that the aggregation of gluten proteins can be the result of competition for water between gluten and DFP during dough mixing [89,90].
Mechanism of interactions between gluten network and the DFP were also studied with application of FT-IR spectroscopy [18]. Analysis of amide I and amide III bands confirmed that the addition of DFP resulted in aggregation of gluten proteins. Aggregated structures contained mainly hydrogen bonded β-sheets and antiparallel-β-sheets formed as a result of interactions between chains of gluten proteins or polysaccharides chains and gluten proteins. The biggest increment of aggregated β-sheets was induced by an increase in the fibre content from 3% to 6%. Moreover, the studies showed that DFP addition induced changes in water populations in the model dough. Changes observed in the OH stretching region as a result of DFP addition suggested that part of water molecules, which are involved in formation of weak hydrogen bonds with proteins in the control sample, can also participate in formation of weak hydrogen bonds with polysaccharide chains. The rest of water molecules induce formation of strong H-bonds with the gluten proteins which can lead to stiffening of the gluten network. However, the results suggested that strong hydrogen bonds are necessary to form gluten network of adequate mechanical properties. The FT-IR spectroscopy was also used to investigate effect of wheat dietary fibre and ferulic acid on the gluten proteins aggregation [91]. The spectra in the amide I band showed that the gluten secondary structure in all systems was dominated by β-sheets, which are considered as the most stable protein conformation. In the case of samples with ferulic acid combined with dietary fibre, a decrease in the number of β-sheets was observed as the amount of acid and fibre increased.
Pomaces obtained after cold pressing oil production can be also considered as DFP, since they are rich in dietary fibre, polyphenols and fatty acids. Effect of the supplementation of the model dough with five oil pomaces from black seed, pumpkin, hemp, milk thistle and primrose on the gluten structure was studied by Rumińska et al. [92] with application of FT-IR and FT-Raman spectroscopy. Analysis of the spectroscopic results indicated that the observed changes depended mainly on the type and amount of fatty acids present in the pomaces although pomaces contained considerable amount of dietary fibre. If the pomaces contain a low number of fatty acids, aggregated β-sheets with intermolecular hydrogen bonds were formed from β-turns and antiparallel-β-sheets. Whereas non-aggregated β-structures were observed for pomaces with a high number of fatty acids.”
Reviewer 2 Report
I believe that the article entitled Spectroscopic methods as analytical tools to determine changes in the structure of gluten, gliadins and glutenins is an interesting one by it subject. The article is a very complex one presenting also different factors that may influence gluten behavior and it impact on final product quality. Some remarks:
- To the dough mixing which is a very important factor the authors presents very few informations regarding the mixing speed effect on gluten compounds which in my opinion is a crucial factor in bread making quality. Please describe in a more extensive way this aspects (for example the effect of mixing speed at different rpm such as 80 rpm, 250 rpm and so on); also the effect of time on mixing speed should be presented;
- The authors describe dough improving effects but only regarding different types of salts or emulsifiers effect. In my opinion should be discussed also the effect of some oxidizing (such as ascorbic acid, glucose oxidase) and reduction agents effect (very used in bread making) on gluten structure;
Author Response
ANSWERS TO THE COMMENTS OF REVIEWER 2
We would like to thank the Reviewer for valuable comments which have helped us to improve quality of our work. The answers to the Reviewer’s specific comments are listed below:
- To the dough mixing which is a very important factor the authors presents very few information regarding the mixing speed effect on gluten compounds which in my opinion is a crucial factor in bread making quality. Please describe in a more extensive way this aspects (for example the effect of mixing speed at different rpm such as 80 rpm, 250 rpm and so on); also the effect of time on mixing speed should be presented.
We agree with the Reviewer that both dough mixing speed and time of dough mixing are very important factors for the dough mixing process because they affect dough microstructure, chemical composition and viscoelastic properties.
As for the effect of dough mixing time, it has been described in the manuscript (L. 251-257): “Ait Kaddour et al. [68] observed changes in the secondary structure of gluten during 20-minute mixing as increase or decrease in the band intensities corresponding to the secondary structures in the amide III band. Increasing mixing time led to increase in the amount of α-helix, β-sheets and β-turns. The results indicated formation of more ordered gluten structure during mixing. Similar results obtained Seabourn et al. [69] who analysed second derivative band area instead of band intensity.”
In the case of mixing speed, we did not find any scientific reports in which spectroscopic methods were used. However, there are a lot of scientific reports concerning influence of speed of dough mixing and dough mixing time on dough rheological properties, dough microstructure, gluten biochemistry in which rheological, electrophoretic, chromatographic and microscopic methods were used. Below we present some references concerned such studies:
- Létang, C.; Piau, M.; Verdier, C. Characterization of Wheat Flour–Water Doughs. Part I: Rheometry and Microstructure. J. Food Eng. 1999, 41, 121–132.
- Amemiya, J.I.; Menjivar, J.A. Comparison of Small and Large Deformation Measurements to Characterize the Rheology of Wheat Flour Doughs. J. Food Eng. 1992, 16, 91–108.
- Gómez, A.; Ferrero, C.; Calvelo, A.; Añón, M.C.; Puppo, M.C. Effect of Mixing Time on Structural and Rheological Properties of Wheat Flour Dough for Breadmaking. Int. J. Food Prop. 2011, 14, 583–598.
- Abang Zaidel, D.N.; Chin, N.L.; Abdul Rahman, R.; Karim, R. Rheological Characterisation of Gluten from Extensibility Measurement. J. Food Eng. 2008, 86, 549–556.
- Codina, G.; Mironeasa, S. Influence of Mixing Speed on Dough Microstructure and Rheology. Food Technol. Biotechnol. 2013, 51, 509–519.
- Osella, C.A.; Sanchez, H.D.; de la Torre, M.A. Effect of dough water content and mixing conditions on energy imparted to dough and bread quality. Cereal Foods World 2007, DOI: 10.1094/CFW-52-2-0070.
- Zheng, H.; Morgenstern, M.P., Campanella, O.H.; Larsen, N.G. Rheological Properties of Dough During Mechanical Dough Development. J. Cereal Sci. 2000, 32, 293-306.
- Sluková M., Levková J., Michalcová A., Horáčková Š., Skřivan P. Effect of the dough mixing process on the quality of wheat and buckwheat proteins. Czech J. Food Sci. 2017, 35, 522–531.
- Peighambardoust, S.H., Dadpour, M.R., Dokouhaki, M. Application of epifluorescence light microscopy (EFLM) to study the microstructure of wheat dough: a comparison with confocal scanning laser microscopy (CSLM) technique. J. Cereal. Sci. 2010, 51, 21–27.
- Amend, T., Belitz, H.-D. Microstructural studies of gluten and a hypothesis on dough formation. Food Struct. 1991, 10, 277–288.
- Gras, P.W., Carpenter, H.C., Anderssen, R.S. Modelling the developmental rheology of wheat-flour dough using extension tests. J. Cereal. Sci. 2000, 31, 1–13.
- The authors describe dough improving effects but only regarding different types of salts or emulsifiers effect. In my opinion should be discussed also the effect of some oxidizing (such as ascorbic acid, glucose oxidase) and reduction agents effect (very used in bread making) on gluten structure.
We thank Reviewer for the comment. We did not discuss the effects of some oxidizing and/or reducing agents on the gluten structure with application of spectroscopic methods because literature contains very few reports of such studies. However, we found a lot of articles concerning effects of these agents on the dough quality, gluten microstructure or gluten biochemistry, which have been studied with application of rheological, electrophoretic, chromatographic and microscopic methods. Below we present some references concerned such studies:
- Joye, I.J.; Lagrain, B.; Delcour, J.A. Endogenous Redox Agents and Enzymes That Affect Protein Network Formation during Breadmaking – A Review. J. Cereal Sci. 2009, 50, 1–10.
- Steffolani, M.E.; Ribotta, P.D.; Pérez, G.T.; León, A.E. Effect of Glucose Oxidase, Transglutaminase, and Pentosanase on Wheat Proteins: Relationship with Dough Properties and Bread-Making Quality. J. Cereal Sci. 2010, 51, 366–373.
- Toyosaki, T. Effects of Hydroperoxide in Lipid Peroxidation on Dough Fermentation. Food Chem. 2007, 104, 680–685.
- Hayta, M.; Schofield, J.D. Heat and Additive Induced Biochemical Transitions in Gluten from Good and Poor Breadmaking Quality Wheats. J. Cereal Sci. 2004, 40, 245–256.
- Sievert, D.; Sapirstein, H.D.; Bushuk, W. Changes in Electrophoretic Patterns of Acetic Acid-Insoluble Wheat Flour Proteins during Dough Mixing. J. Cereal Sci. 1991, 14, 243–256.
- Aminlari, M.; Majzoobi, B. Effect of Chemical Modification, PH Change, and Freezing on the Rheological, Solubility, and Electrophoretic Pattern of Wheat Flour Proteins. J. Food Sci. 2002, 67, 2502-2506.
- Nakamura, M.; Kurata, T. Effect of L-Ascorbic Acid and Superoxide Anion Radical on the Rheological Properties of Wheat Flour-Water Dough. Cereal Chem. 1997, 74, 651–655.
- Indrani, D.; Rao, G.V. Effect of Additives on Rheological Characteristics and Quality of Wheat Flour Parotta. J. Texture Stud. 2006, 37, 315–338.
- Miller, K.A.; Hoseney, R.C. Effect of Oxidation on the Dynamic Rheological Properties of Wheat Flour-Water Doughs. Cereal Chem. 1999, 76, 100–104.
Round 2
Reviewer 1 Report
I think the changes to the manuscript have helped to make it more focused. There are still some minor concerns left.
1) The title feels somewhat clumsy. Perhaps "Effects of physical and chemical factors on the structure of gluten, gliadins and glutenins as studied with spectroscopic methods" can be an option.
2) There some minor punctuation and formatting results, the list is not exhaustive, so please check more.
Lines 181, 182, 431, 446, 521, 523, 580: the spaces between words are too large at places.
Line 187: redundant dots
Line 295: redundant comma
Line 503: space before dot
Line 668: redundant comma
Line 773: redundant symbols
Line 743: cm-1 "-1" is not in the superscript
Lines 190, 191, 293: -1 is split into the next line from cm.